# An Electrochemical Immunosensor for the Determination of Procalcitonin Using the Gold-Graphene Interdigitated Electrode

**DOI:** 10.3390/bios12100771

**Published:** 2022-09-20

**Authors:** Mahmoud Amouzadeh Tabrizi, Pablo Acedo

**Affiliations:** Electronic Technology Department, Universidad Carlos III de Madrid, 28911 Leganés, Spain

**Keywords:** interdigitated electrode, laser engraved graphene, immunosensor, electrochemistry, procalcitonin

## Abstract

Procalcitonin (PCT) is considered a sepsis and infection biomarker. Herein, an interdigitated electrochemical immunosensor for the determination of PCT has been developed. The interdigitated electrode was made of the laser-engraved graphene electrode decorated with gold (LEGE/Au_nano_). The scanning electron microscopy indicated the LEGE/Au_nano_ has been fabricated successfully. After that, the anti-PTC antibodies were immobilized on the surface of the electrode by using 3-mercaptopropionic acid. The electrochemical performance of the fabricated immunosensor was studied using electrochemical impedance spectroscopy (EIS). The EIS method was used for the determination of PCT in the concentration range of 2.5–800 pg/mL with a limit of detection of 0.36 pg/mL. The effect of several interfering agents such as the C reactive protein (CRP), immunoglobulin G (IgG), and human serum albumin (HSA) was also studied. The fabricated immunosensor had a good selectivity to the PCT. The stability of the immunosensor was also studied for 1 month. The relative standard deviation (RSD) was obtained to be 5.2%.

## 1. Introduction

Procalcitonin (PCT) which is made of 116 amino acids [1] is a potential early biomarker of several infectious diseases such as bacteria, viruses, and fungi in the early stage [2]. A high PCT antigen level (>2.0 µg/L) indicates a high probability of sepsis disease. Sepsis is a fatal response of the immune system to an infection [3]. Hence, it is essential to fabricate a highly sensitive, selective device to detect the PCT antigen at the trace level. Various techniques have been used for the detection of PCT antigen such as chemiluminescence [4], electrochemiluminescence [5], optical fiber [6], and electrochemical [7]. Among them, electrochemical biosensors have various advantages such as high sensitivity, easy use, and portability [8,9,10]. EIS is an electrochemical method that has been widely used for the detection of biomarkers [11,12]. Nowadays, thanks to nanomaterials, the analytical performances of electrochemical biosensors improve dramatically [13,14]. Graphene is a 2-dimensional carbon-based nanomaterial have been used wildly for the fabrication of electrochemical biosensors [15,16]. Up to now, several methods have been used for the fabrication of graphene [17,18]. The laser engraved Kapton is an eco-friendly, fast, and cheap method that allows us to design a graphene-based electrode not only in any shape but also on a high scale [19]. During the process of fabrication, the Kapton, which is a polyimide film burns by a laser beam, inducing sp^3^-to-sp^2^ conversion [20]. The generated material has most of the properties of graphene. The laser engraved graphene-based electrodes (LEGEs) have been used for the fabrication of electrochemical biosensors [21,22,23]. The LEGEs have not only highly conductive, and high surface area electrodes but also, are flexible, and cheap.

In this research work, we designed label-free electrochemical impedance spectroscopy (EIS) based immunosensor for PTC detection using the interdigitated graphene–gold electrode. The gold nanoparticles were deposited on the surface of the interdigitated electrode electrochemically to attach the anti-PTC antibodies on the electrode surface via 3-mercaptopropionic acid (3-MPA). The results indicated that in the presence of the different concentrations of PTC antigen, the electron transfer resistance (R_et_) of the biosensor increased due to the interaction of the anti-PTC antibodies with PTC antigen. The proposed biosensor had good selectivity, sensitivity, and stability.

## 2. Materials and Methods

### 2.1. Reagents and Chemicals

All chemicals were of analytical reagent grade and used without further purification. Double deionized (DI) water (18.6 MΩ) was used throughout the research work. Sodium tetrachloroaurate (NaAuCl_4_), phosphoric acid (H_3_PO_4_), potassium chloride (KCl), potassium hydroxide (KOH), potassium ferricyanide (Fe(CN)_6_^3−^), potassium ferrocyanide (Fe(CN)_6_^4−^), and chitosan were obtained from Alfa Aesar. Anti-PCT antibody and PCT antigen were purchased from Abcam (Cambridge, United Kingdom). Human serum albumin (HSA), human immunoglobulin G (HIgG) C-reactive protein (CRP) antigens, N-(3-dimethyl aminopropyl)-N-ethyl carbodiimide (EDC), N-Hydroxysulfosuccinimide (sulfo-NHS), bovine serum albumin (BSA), and 3-MPA were obtained from Sigma-Aldrich (St. Louis, MO, USA).

### 2.2. Apparatus

The morphology of the electrode surface was investigated using scanning electron microscopy (SEM) (Field Electron and Ion (FEI)) (FEI, Hillsboro, OR, USA). The elemental analysis was performed using an energy dispersive analysis of X-rays (EDX) (EDAX, Mahwah, NJ, USA). The electrochemical reduction of gold ions to gold nanoparticles was performed using a potentiostat from Metrohm-DropSens Model µStat300. Electrochemical impedance spectroscopy (EIS) data were performed using an ISX-3 impedance analyzer (Sciospec, Bennewitz, Germany) in phosphate-buffered saline (0.1 M PBS, pH 7.4) containing 5.0 mM Fe(CN)_6_^4−/3−^ couple (1:1) as the redox probe. The reference and counter electrodes were made of graphene. The interdigitated electrodes (sensing electrodes) were made of graphene–gold.

### 2.3. Fabrication of the Immunosensor

The fabrication of the interdigitated graphene electrode (IDGE) was completed by carbonization of the Kapton tape with a diode laser beam. To do that, 30% power of the laser was applied for the fabrication. As shown in Appendix A, there are two interdigitated electrodes, a reference electrode, and a counter electrode. After that, the IDGE was raised with DI water to remove any organic pollution components that were produced during the carbonization of Kapton. Before electrodeposition of the gold nanoparticles, two interdigitated electrodes were connected to each other with copper tape to decorate them with gold nanoparticles simultaneously. Subsequently, 500 µL of 0.1 mM NaAuCl_4_ was dropped on the surface of the IDGE and then applied to −0.8 V for 1 min. During this process, the gold ions were reduced to gold nanoparticles and decorated the surface of the IDGE (IDGE/Au_nano_). After that, the electrode was washed for 1 min with running DI water. To functionalize the surface of the IDGE/Au_nano_ with carboxylic acid groups, 5 µL of 3-MPA (1 mM, pH 7.4) was dropped on the electrode surface for overnight at 5 °C. The IDGE/Au_nano_ modified with 3-MPA (IDGE/Au_nano_/3-MPA) was washed with 0.1 M PBS (pH 7.4) for 1 min. To immobilize the anti-PCT antibodies, the carboxylic acid groups on the surface of IDGE/Au_nano_/3-MPA should first activate. To do that, 500 µL of EDC/Sulfo-NHS (10 mM/20 mM, pH 7.4) was drooped on the surface of IDGE/Au_nano_/3-MPA and allowed to interact with carboxylic acid groups for 1 h. Later, the electrode was rinsed with 0.1 M PBS quickly and 500 µL of 1 µg/mL anti-PTC antibody was dropped on the electrode surface. After 2 h, the electrode was washed with 0.1 M PBS and 500 µL of 1 µg/mL BSA solution (pH 7.4) was dropped on it to block the activated sites on the surface of the electrode to avoid any nonspecific interaction between the biomolecules and the active sites. Finally, the electrode was washed with 0.1 M PBS and stored at 5 °C. The proposed immunosensor was named (IDGE/Au_nano_/3-MPA/Anti-PTC/BSA). The fabricated interdigitated immunosensor was stored at 4 °C in a refrigerator inside a dry and dark box when not in use. The photo image of the electrochemical setup is shown in Appendix A.

### 2.4. Measurement Procedure of PCT

Before the measurement, 100 µL of human plasma sample and 100 µL of 0.2 M PBS solution containing various concentrations of PCT were mixed for 2 min with a mini mixer. Then, the mixture was dropped on the surface of IDGE/Au_nano_/3-MPA/Anti-PTC/BSA for 60 min at 4 °C to incubate the PCT antigen with the anti-PCT antibodies of the immunosensor. After that, the immunosensor was washed with 0.1 M PBS for 1 min. The EIS method in the form of the Nyquist plot was applied for PCT antigen detection in 0.1 M PBS (pH 7.4) containing 5 mM Fe(CN)_6_^3−/4−^ redox probe. The stepwise fabrication of the immunosensor and its response mechanism is shown schematically in Figure 1.

## 3. Results

### 3.1. The Surface Characterization of the Electrodes

Figure 2 shows the surface morphology of the IDGE/Au_nano_. As shown in Figure 2A, the fingers of the IDGE were separately well and the average width of the fingers and distance between them were 330 ± 2 µm, and 148 ± 3 µm, respectively. The medium magnification image of the IDGE/Au_nano_ showed the fingers had a uniform porous structure (Figure 2B). Finally, the SEM image of the IDGE/Au_nano_ in the high magnification was recorded. As can be seen in Figure 2C, the graphene was decorated with gold nanoparticles. The average diameter size of the gold nanoparticles was 7 ± 0.6 nm (n = 10).

In addition, the elemental analysis of the IDGE/Au_nano_/3-MPA/Anti-PTC/BSA was performed with EDX (Appendix A). As can be seen, a big carbon element peak related to graphene and/or Kapton backbone, a big peak of oxygen element related to Kapton backbone and/or anti-PCT antibodies, a medium peak of nitrogen element to Kapton backbone and/or anti-PCT antibodies, Au element related to gold nanoparticles, a sulfur element related to the thiol group of 3-MPA are clearly seen. It indicates the gold nanoparticles and 3-MPA were coated on the surface of the electrode. Since, the N, O, and C elements could be found in the both Kapton backbone and antibody, we cannot claim that the antibody was immobilized on the electrode surface based on the EDX data. To solve this problem, we also used the EIS and cyclic voltammetry (CV) methods for the characterization of the biosensor.

Figure 3 shows the Nyquist plots of the IDGE (a), IDGE/Au_nano_ (b), IDGE/Au_nano_/3-MPA (c), IDGE/Au_nano_/3-MPA/Anti-PCT (d), IDGE/Au_nano_/3-MPA/Anti-PCT/BSA (e), IDGE/Au_nano_/3-MPA/Anti-PCT/BSA/PCT (f) in 5 mM Fe(CN)_6_^3−^/^4−^ couple (1:1) and 0.1 M PBS. The Nyquist plot includes a semicircular portion at higher frequencies related to the electron-transfer-limited process and a linear part at the lower frequency range related to the diffusion-limited process. The diameter of the semicircle equals the electron-transfer resistance (R_et_). Zero potential was applied for the EIS measurement because the formal potential of the Fe(CN)_6_^3−^/^4−^ redox probe was zero.

As shown, gold nanoparticles decrease the electron transfer resistance (R_et_) of the IDGE (a) from 2168 Ω to 1919 Ω, indicating the gold nanoparticles facilitated the rate of electron transfer of the Fe(CN)_6_^3−^/^4−^ redox probe. After the functionalization of the electrode with 3-MPA, the R_et_ increased to 2377 Ω, due to the electrostatic repulsion interaction between the negatively charged redox probe and carboxylic acid groups on the IDGE/Au_nano_/3-MPA. The immobilization of the anti-PCT antibodies increased the R_et_ of the electrode to 2609 Ω, because of the mass transfer limitation of the Fe(CN)_6_^3−^/^4−^ redox probe to the electrode surface. Blocking the active sites with BSA, the R_et_ of the electrode increased to 2744 Ω. As 10 pg/mL PCT antigen was dropped on the immunosensor surface to interact with the immobilized antibodies, the R_et_ of the electrode increased to 2993 Ω, indicating the mass transfer limitation of Fe(CN)_6_^3−^/^4−^ redox probe to the electrode surface increased. This demonstrates that the step-wise modification process was prepared successfully. Additionally, Appendix A shows the CV of the electrodes after each step modification. As shown, the signal of the graphene electrode increased as gold nanoparticles were deposited on the surface of the graphene electrode. After that, any change on the surface of the electrode decreased the CV signal, verifying the results obtained by EIS.

### 3.2. Optimization of the Parameters

The effect of the anti-PCT antibodies value and incubation time on the response of the IDGE/Au_nano_/3-MPA/Anti-PCT/BSA to 800 pg/mL PCT antigen were investigated. As shown in Figure 4A, the R_et_ of the immunosensor increased as the anti-PCT antibody concentration increased from 0.25 µg/mL to 1.0 µg/mL and after that the signal remained constant, indicating the maximum value of the anti-PCT antibodies could be immobilized on the electrode surface was 1.0 µg/mL. The effect of the incubation time on the response of the immunosensor was also studied (Figure 4B). As can be seen, the response of the immunosensor to 800 pg/mL PCT antigen kept increasing up to 60 min and did not change after that, indicating the interaction of anti-PCT antibodies and PCT antigens reached a steady state. Therefore, 1.0 µg/mL of anti-PCT antibodies was used for the fabrication of the immunosensor, and 60 min was selected as the optimized incubation time between the anti-PCT antibody–PCT antigen during the investigation.

### 3.3. Electrochemical Measurement and Analysis of the Results

Figure 5A shows the Nyquist plots of the IDGE/Au_nano_/3-MPA/Anti-PCT/BSA in a 0.5 mM Fe(CN)_6_^3−^/^4−^ solution after the incubation with different concentrations of PCT antigen. As shown, as the PCT antigen concentration was increased from 2.5–800 pg/mL, the R_et_ of the IDGE/Au_nano_/3-MPA/Anti-PCT/BSA increased. Figure 5B shows that the R_et_ has a linear relationship with the logarithm of the PCT antigen concentration with a regression equation below:ΔR_et_ (kΩ) = 1.386 Log C_PCT_ (pg/mL) + 1.937(1)

The limit of detection (LOD) was calculated to be 0.36 pg/mL at (3σ/S), where σ is the standard deviation of the blank signal for five different IDGE/Au_nano_/3-MPA/Anti-PCT/BSA, and S is the slope of the calibration curve.

The Langmuir isotherm constant (K_L_), dissociation constant (K_d_), and the maximum number of binding sites (R_et max_) for the IDGE/Au_nano_/3-MPA/Anti-PCT/BSA were calculated by using the equation below (Figure 5C) [24,25]: (2)$$\frac{C_{\mathrm{PCT}}}{R_{\mathrm{et}}}=\frac{1}{K_{L}\times R_{\mathrm{et}\;\max}}+\frac{C_{\mathrm{PCT}}}{R_{\mathrm{et}\;\max}}$$
where R_et max_ is the steady-state signal after the addition of a biomarker.

The values of 1/R_et max_ and 1/K_L_×R_et max_ can be obtained from the slope and the intercept point of Figure 5C, respectively. The values of R_et max_, K_L_, and K_d_ (1/K_L_) were calculated to be 6.17 kΩ, 0.057 (pg/mL)^−1^, and17.57 pg/mL (3.8 pM), respectively. The obtained K_d_ value for IDGE/Au_nano_/3-MPA/Anti-PCT/BSA was lower than the peptide-based aptasensor 0.39 nM [26], indicating the proposed immunosensor had a better affinity to PCT antigen.

Additionally, the Gibbs free energy of desorption bind for the PCT antigen/Anti-PCT antibody was calculated to be −50.8 kJ/mol using Equation (3) [27]:ΔG = 2.03 × R × T × log (K_d_)(3)
where R is the universal gas constant of 8.31 J/(K×mol), and T is the temperature in Kelvin (298.15 K).

### 3.4. Stability, Reproducibility, and Selectivity of the IDGE/Au_nano_/3-MPA/Anti-PCT/BSA

The stability of the IDGE/Au_nano_/3-MPA/Anti-PCT/BSA was studied. The relative standard deviation (RSD) of R_et_ of immunosensor was changed by approximately 5.2% after 1 month. It demonstrates that the IDGE/Au_nano_/3-MPA/Anti-PCT/BSA has good stability. The reproducibility of the immunosensorwas also studied for measurement of 500.0 pg/mL of PCT antigen with four different immunosensors. The RSD was calculated as 6.8%.

The selectivity of the immunosensor was also investigated. For this purpose, different IDGE/Au_nano_/3-MPA/Anti-PCT/BSA were fabricated to measure 10 pg/mL PCT, and the interfering agents (10.0 pg/mL CRP, HSA, and HIgG) with the different IDGE/Au_nano_/3-MPA/Anti-PCT/BSA (Appendix A). The selectivity factor (α) was calculated by using the equation below [28]:(4)$$\alpha =\frac{(\Delta R_{et})TM\;}{(\Delta R_{et})IA\;}$$
where (ΔR_et_)_TM_ and (ΔR_et_)_IA_ are the changes in the response of the immunosensor in the absence and presence of 10 pg/mL PCT and 10 pg/mL interfering agents, respectively. The value of α for the immunosensor to CPT in the presence of CRP, HSA, and HIgG was 5.4, 8.3, and 8.1, respectively. Since the values of α were bigger than 1, the selectivity of the IDGE/Au_nano_/3-MPA/Anti-PCT/BSA to PCT antigen was high.

The analytical performances of the IDGE/Au_nano_/3-MPA/Anti-PCT/BSA were compared with the other biosensors for PCT (Table 1). As can be seen, the analytical performances of the proposed immunosensor were better than theirs. Although the LOD of the GCE/NiFe PBA nanocubes@TB/Anti-PCT is a little lower than the proposed immunosensor [29], its linear range of it is narrow.

## 4. Conclusions

In this paper, the interdigitated graphene–gold nanoparticles-based electrode was fabricated by engraving Kapton tape with a diode laser beam and then applied for the fabrication of the immunosensor for PCT in the human plasma sample. The EIS method was used for the detection of the PCT concentration in the range of 2.5–800 pg/mL. The limit of detection was found to be 0.36 pg/mL. Additionally, the result indicated the K_d_ of the proposed immunosensor was lower than the peptide-based aptasensor indicating the high affinity of the bio-recognizers (the immobilized anti-PCT antibodies on the immunosensor) to PCT in comparison with peptide-based bio-recognizer. Furthermore, the proposed immunosensor showed high selectivity to PCT in the presence of CRP, IgG, and HSA. The analytical performances of the IDGE/Au_nano_/3-MPA/Anti-PCT/BSA were better the most of the biosensors in Table 1. The proposed immunosensor had high stability, making it a good candidate for the detection of PCT in real samples. Although the IDGE/Au_nano_/3-MPA/Anti-PCT/BSA for PCT showed good sensitivity and selectivity, it has a disadvantage. The EIS is an expensive technique to be used for the fabrication of a point of care sensor for PCT compare to the potentiostat methods such as differential pulse voltammetry. Therefore, in the future, we will work on this issue to fabricate a cheap immunosensor.

## Figures and Tables

**Figure 1 biosensors-12-00771-f001:**
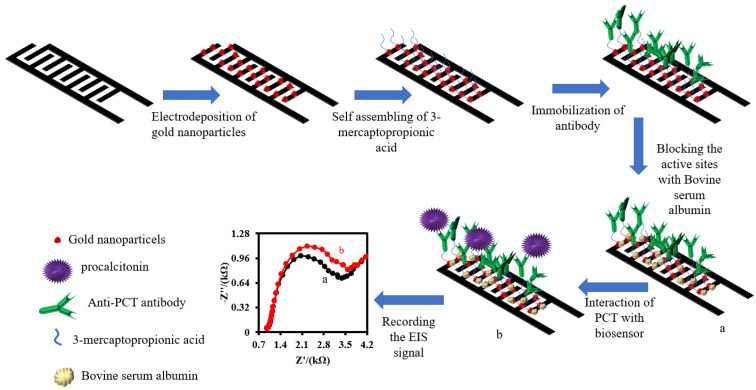
Schematic illustration of the fabrication of IDGE/Au_nano_/3-MPA/Anti-PTC/BSA and response mechanism for PTC antigen.

**Figure 2 biosensors-12-00771-f002:**
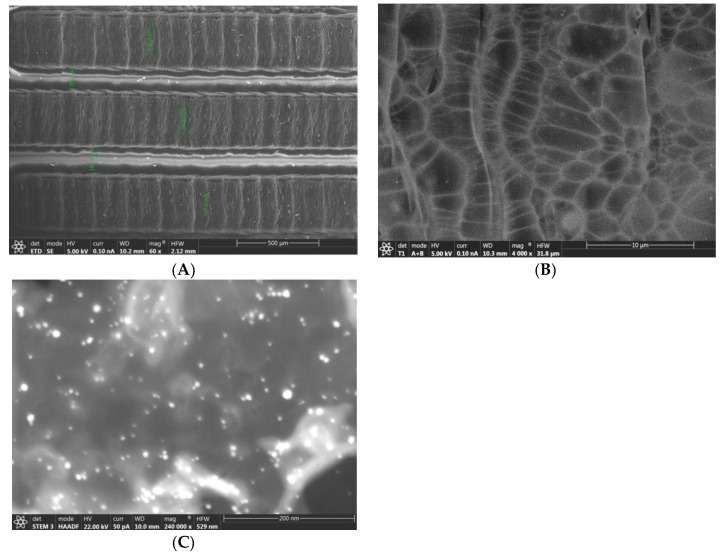
(**A**–**C**) SEM images of the IDGE/Au_nano_ in different magnifications.

**Figure 3 biosensors-12-00771-f003:**
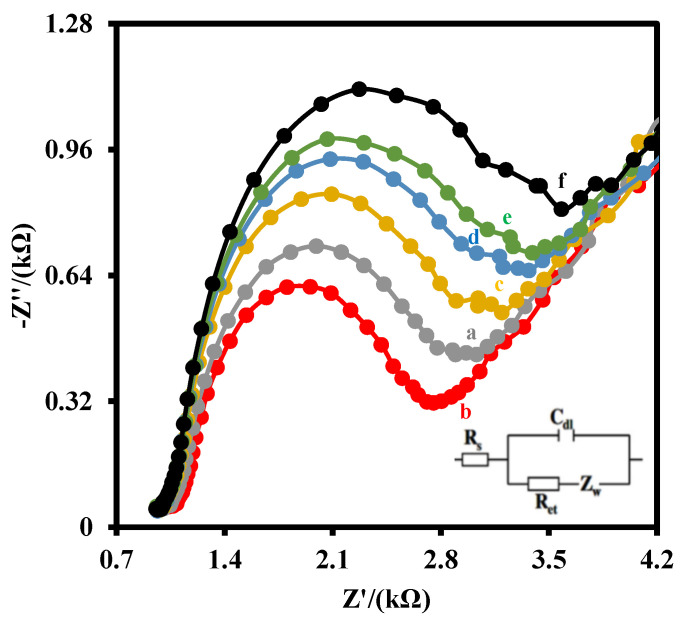
Nyquist plots of the EIS of the IDGE (a), IDGE/Au_nano_ (b), IDGE/Au_nano_/3-MPA (c), IDGE/Au_nano_/3-MPA/Anti-PCT (d), IDGE/Au_nano_/3-MPA/Anti-PCT/BSA (e), IDGE/Au_nano_/3-MPA/Anti-PCT/BSA/PCT (10 pg/mL) (f) in 5 mM Fe(CN)_6_^3−^/^4−^ couple (1:1) and 0.1 M PBS. The inset is the equivalent electric circuit compatible with the Nyquist diagrams. R_s_: solution resistance, R_et_: electron transfer resistance, C_dl_: double layer capacitance, Z_w_: Warburg impedance. AC amplitude voltage was 10 mV, DC voltage was 0 V, and frequency range was 100,000–0.1 Hz.

**Figure 4 biosensors-12-00771-f004:**
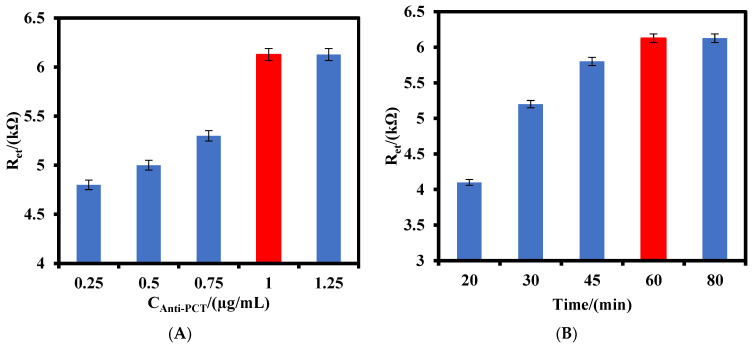
(**A**) Effect of anti-PCT antibody value and (**B**) the incubation time on the response of the IDGE/Au_nano_/3-MPA/Anti-PCT/BSA to 800 pg/mL PCT antigen.

**Figure 5 biosensors-12-00771-f005:**
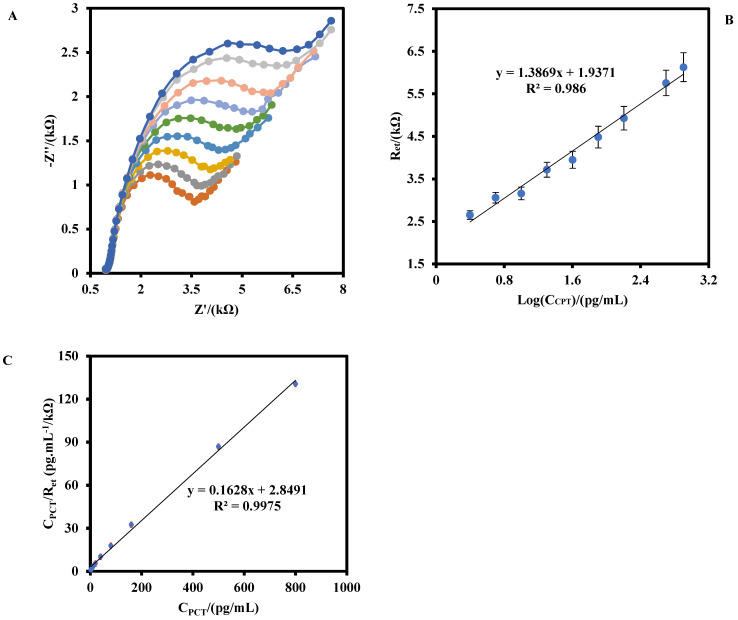
(**A**) EIS response of IDGE/Au_nano_/3-MPA/Anti-PCT/BSA at the optimum operating conditions for different of PCT concentrations (2.5, 5.0, 10.0, 20.0, 40.0, 80.0, 160.0, 500.0, and 800.0 pg/mL). (**B**) The corresponding linear logarithmic calibration plot of the immunosensor. (**C**) The plat of Langmuir binding isotherm model for IDGE/Au_nano_/3-MPA/Anti-PCT/BSA. The error bars were related to the five different immunosensors.

**Table 1 biosensors-12-00771-t001:** Comparison of the analytical performance of the IDGE/Au_nano_/3-MPA/Anti-PCT/BSA with some of the biosensors for PCT antigen.

Biosensor	Method	Linear Range	LOD	Time	Ref
GCE/NiFe PBA nanocubes@ TB/Anti-PCT	DPV	0.001–25 ng/mL	0.3 pg/mL	60 min	[8]
Fused silica capillary/Anti-PCT and biotinylated HRP/streptavidin Anti-PCT antibody	CL	2.5–8.0 × 10^4^ pg/mL	0.5 pg/mL	60 min + 60 min	[4]
GE/DSP/Anti-PCT	EIS	0.01–10 ng/mL	0.1 ng/mL	90 min	[30]
GCE/Gr-CHIT/Anti-PCT and CdTe/Silica_nano_/Anti-PCT	ECL	0.01–20 ng/mL	3.4 pg/mL	45 + 45 min	[31]
GE/peptide	EIS	12.5–250 ng/mL	12.5 ng/mL	-	[26]
ITO/WO3/NCQDs/Sb_2_S_3_/PDA/Anti-PCT/BSA	PEC	0.001–100 ng/mL	0.42 pg/mL	-	[10]
IDGE/Au_nano_/3-MPA/Anti-PCT/BSA	EIS	2.5–800 pg/mL	0.36 pg/mL	60 min	This work

GCE: Glassy carbon electrode; NiFe PBA nanocubes@TB: Toluidine blue functionalized NiFe Prussian-blue analog nanocubes; PANI NRs: Polyaniline nanorod arrays; rGO-Au: Reduced graphene oxide-gold nanoparticles; DSP: dithio bis(succinimidyl propionate); PEC: Photoelectrochemical; ITO: Indium tin oxide; ECL: Electrogenerated chemiluminescence; Gr-CHIT: Graphene–chitosane; GE: Gold electrode; WO_3_/NCQDs/Sb2S3/PDA/Anti-PCT: Tungsten trioxide/nitrogen-doped carbon quantum dots/antimony trisulfide/polydopamine.

## Data Availability

Appendix A associated with this article can be found in the online version including Appendix A.

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
