# Peer review of "An Electrochemical Immunosensor for the Determination of Procalcitonin Using the Gold-Graphene Interdigitated Electrode"

_biosensors, 2022, doi:10.3390/bios12100771_

Round 1

Reviewer 1 Report

In this work, M.A. Tabrizi et al. have developed an electrochemical immunosensor for the detection of procalcitonin, a sepsis biomarker. The authors fabricated interdigitated electrodes by engraving Kapton with a diode laser beam. For the sensor, they modified the electrode with gold nanoparticles and used the EDC/NHS reaction to activate the 3-mercaptopropionic acid for the immobilization of the antibody. They studied the response of the sensor using EIS, and they also characterized the selectivity and stability of the sensor. I would recommend its publication in Biosensors after some minor comments:

1.      Regarding the stability assay, how did the authors perform it? I think they should explain how they store the electrodes. Did they follow a specific protocol? Were the electrodes stored under dry or wet conditions?

2.      They express the selectivity with the selectivity factor. I think it would be useful to also have a figure with the values of Rct for each interfering agents and PCT. 

Author Response

Dear Referee,

Thank you very much for concerning our manuscript “biosensors-1928966”. We are highly thankful to you for having given us valuable suggestions for the improvement of our manuscript; they are helpful indeed. We accepted all comments and submitted a list of the changes made to the manuscript, following your comments point by point (see next page). With this, we hope that this study now meets the criteria for publication in Biosensors.

Yours sincerely,

Comments

In this work, M.A. Tabrizi et al. have developed an electrochemical immunosensor for the detection of procalcitonin, a sepsis biomarker. The authors fabricated interdigitated electrodes by engraving Kapton with a diode laser beam. For the sensor, they modified the electrode with gold nanoparticles and used the EDC/NHS reaction to activate the 3-mercaptopropionic acid for the immobilization of the antibody. They studied the response of the sensor using EIS, and they also characterized the selectivity and stability of the sensor. I would recommend its publication in Biosensors after some minor comments:

  1. Regarding the stability assay, how did the authors perform it? I think they should explain how they store the electrodes. Did they follow a specific protocol? Were the electrodes stored under dry or wet conditions?

 Response: It was done. Kindly see page 3, lines 100-101.

  1. They express the selectivity with the selectivity factor. I think it would be useful to also have a figure with the values of Rctfor each interfering agent and PCT. 

Response: It was done. Please see the supplementary data file, Fig.S3.

Thank you for taking the time to review our work.

Reviewer 2 Report

In this paper, gold-graphene interdigitated electrode was used to develop an electrochemical immunosensor for the determination of procalcitonin as a sepsis biomarker. The paper is not well presented or written. More accurate interpretation of the results is required. Required to clarify many ambiguities in the results. Therefore, the paper required thorough major revision before publication in this journal.

 Comments:

1.      The abstract should be rewrite. It is written like a methodology.

2.      The effect 20 of------ human serum albumin (HSA). What next? You mean tested or performed? The sentence should be revised

3.      In introduction, it should be nanomaterials not nonmaterial.

4.      In 3.1 section, 330±2, and 148±3. what unit it is? Is it micro or nano meter?

5.      7 nm±6.6.? Looks like something is strange. Actual size and the standard deviation both are almost same. How was the average size of the particle calculated? Provided specific data.

 6.      There is no explanation regarding the mentioned electrochemical equivalent circuit in Fig. 3.

 7.      In section 3.3, the tested PCT concentration is 10-1000 pg/mL whereas in Fig. 5A and in caption also the PCT concentration is mentioned as 2.5 -800 pg/mL?

 8.      The limitation of the proposed sensor and how it can be overcome the limitation in future research should be mentioned.

 9.      English is very poor. Whole manuscript needs to check for the English errors. It is recommended to consult native English speaker for the English corrections.

Author Response

Dear Referee,

Thank you very much for concerning our manuscript “biosensors-1928966”. We are highly thankful to you for having given us valuable suggestions for the improvement of our manuscript; they are helpful indeed. We accepted all comments and submitted a list of the changes made to the manuscript, following your comments point by point (see next page). With this, we hope that this study now meets the criteria for publication in Biosensors.

Yours sincerely,

Comments

In this paper, gold-graphene interdigitated electrode was used to develop an electrochemical immunosensor for the determination of procalcitonin as a sepsis biomarker. The paper is not well presented or written. More accurate interpretation of the results is required. Required to clarify many ambiguities in the results. Therefore, the paper required thorough major revision before publication in this journal.

 Comments:

  1. The abstract should be rewritten. It is written like a methodology.

Response: It was rewritten.

  1. The effect 20 of------ human serum albumin (HSA). What next? You mean tested or performed? The sentence should be revised

Response: It was corrected. Kindly see page 1, line 17.

  1. In introduction, it should be nanomaterials not nonmaterial.

Response: It was corrected. Kindly see page 1, line 35.

  1. In 3.1 section, 330±2, and 148±3. what unit it is? Is it micro or nano meter?

Response: it was micromiter and we wrote it. We forgot. Sorry about it.

  1. 7 ±6.6 nm.? Looks like something is strange. Actual size and the standard deviation both are almost same. How was the average size of the particle calculated? Provided specific data.

Response: it was corrected to 7nm±0.6. It was a typo error. The RSD was calculated from the size of 10 nanoparticles.

  1. There is no explanation regarding the mentioned electrochemical equivalent circuit in Fig. 3.

Response: it was done. Kindly see page 5, lines 143-146.

  1. In section 3.3, the tested PCT concentration is 10-1000 pg/mL whereas in Fig. 5A and in caption also the PCT concentration is mentioned as 2.5 -800 pg/mL?

Response: It was corrected to2.5 -800 pg/mL. Please see page 7, line 198.

  1. The limitation of the proposed sensor and how it can be overcome the limitation in future research should be mentioned.

Response: It was done. Page 10 lines 274-278.

  1. English is very poor. Whole manuscript needs to check for the English errors. It is recommended to consult native English speaker for the English corrections.

Response: Regarding your comment, we have revised the whole manuscript carefully and tried to avoid any grammar or syntax errors. In addition, we have asked several colleagues who are skilled authors of English language papers to check the English.

Thank you for taking the time to review our work.

Round 2

Reviewer 2 Report

The authors have improved the manuscript quality in the revised manuscript. Also, provided the response to reviewer's comments. Therefore the paper can be published in the journal.

Author Response

Thank you very much.

Sincerely